# Characteristics of a Danish Post-COVID Cohort Referred for Examination due to Persistent Symptoms Six Months after Mild Acute COVID-19

**DOI:** 10.3390/jcm11247338

**Published:** 2022-12-10

**Authors:** Jane Agergaard, William M. Ullahammer, Jesper D. Gunst, Lars Østergaard, Berit Schiøttz-Christensen

**Affiliations:** 1Department of Infectious Diseases, Aarhus University Hospital, 8200 Aarhus, Denmark; 2Department of Clinical Medicine, Aarhus University, 8000 Aarhus, Denmark; 3Research Unit of General Practice, University of Southern Denmark, 5230 Odense, Denmark

**Keywords:** long COVID, post-COVID-19 condition, SARS-CoV-2, post-COVID clinic, post-COVID symptom questionnaire

## Abstract

Background: Post-COVID Clinics were recommended for patients with persistent symptoms following COVID-19, but no specific tests were suggested for evaluation. This study aimed to present a post-COVID clinic patient cohort and evaluate the use of a post-COVID symptom questionnaire (PCQ) score. Methods: Patients were referred from a population of approximately 1 million citizens. PCQ and standardized health scales were registered. Descriptive analyses were performed to assess the prevalence of symptoms, and correlation analyses was undertaken to asses convergent and discriminant trends between PCQ scores and health scales. Results: Of 547 patients, 447 accepted inclusion. The median age was 47 years and 12% of the patients were hospitalized. At a median of 6.3 (IQR 4.4–9.9) months after the onset of symptoms, 82% of the patients reported both physical exhaustion and concentration difficulties. Functional disability and extreme fatigue were reported as moderate to severe by 33% and 62% of the patients, respectively. The PCQ score correlated significantly with each of the standardized health scales. Conclusion: Patients referred to a Post-COVID Clinic were previously generally healthy. At the time of diagnosis, they reported multiple symptoms with severely affected health. The PCQ score could be used as valid measure of Post-COVID severity.

## 1. Introduction

Long-term symptoms following coronavirus disease (COVID-19) (long COVID) were recognized early in the epidemic [1,2,3] and have been documented in various cohorts [4,5,6,7,8,9]. A number of symptoms have been described following hospitalization with moderate or severe acute COVID-19 [10,11], as well as mild acute COVID-19 [4,8,9,12]. Fatigue, dyspnea, altered smell and taste, concentration difficulties and muscle aches/reduced strength are among the most frequently reported symptoms [6,8,9,12,13].

The World Health Organization (WHO) published a clinical case definition of post-COVID condition [14], but there is still no diagnostic criteria to define patients with a severe or long duration of long COVID. Furthermore, the time interval from COVID-19 to long COVID and the symptoms of long COVID vary between guidelines [14,15,16]. Core outcome sets (COS) have been suggested to include *“functioning, symptoms and conditions”* of specific areas/organ systems (cardiovascular, respiratory nervous system, cognitive, mental and physical) and to include recovery. *“Muscle and joint symptoms and conditions*” did not meet an a priori consensus criteria for inclusion in the COS. However, these were rated of high importance among people with long COVID [17].

In Denmark, concerns of how to care for patients with long COVID led to the establishment of national post-COVID-19 services in the Departments of Pulmonology and Infectious Diseases. During the summer of 2020, the Danish Health Authority gathered experts to draft recommendations. In October 2020, it was recommended to establish post-COVID clinics in all of the five Danish regions to care for patients with this poorly defined disease, and to gather data for quality assurance and research [15]. Multiple symptoms without obvious explanations were reported and the clinics were recommended to establish multidisciplinary clinical teams in close relation to research initiatives. Post-COVID services improve clinical expertise and the care for the individual long COVID patient [15,16], and the cohorts established in these settings might be the key to support detection of a pathophysiological cause of long COVID. Based on observations, long COVID might affect 50% of patients with SARS-CoV-2 [10], but some symptoms improve spontaneously [7,10,18]. Post-COVID clinics were reserved for patients with unexpected, complex and prolonged symptoms persisting for more than 12 weeks after infection with SARS-CoV-2. This was in line with the international definition of post-COVID-19 condition and recommendations for the clinical approach [16,19]. Now, 2½ years after the first case of COVID-19, long-term follow-up studies report persistent debilitating disease [10,18], which may constitute a huge challenge for the health care system. All available data characterizing severely affected individuals are thus needed.

The objective of this study was to describe patients included during the first nine months in a post COVID clinic in a Danish region, Central Denmark Region (approx. 1.3 million inhabitants), using a post-COVID-19 symptom questionnaire (PCQ) and standardized health scale measures, and to compare PCQ scores with standardized health scales. We hypothesized that the severity of symptoms reported in the PCQ correlate to the severity measured by other standardized measures. This is the first description of a Danish post-COVID cohort including all patients referred for examination in a regional post-COVID-19 clinic due to persistent symptoms.

## 2. Materials and Methods

In the beginning of 2021, a post-COVID clinic was established and a multidisciplinary research project on long-term health consequences of COVID-19 illness (MULTICOV), registering patient symptoms was initiated at Aarhus University Hospital in Central Denmark Region, covering an uptake area with 17% (1.0 of 5.8 million) of the Danish Population.

### 2.1. Data Collection

As early reports described possible long-term cognitive disturbances, dyspnea and fatigue and a number of other less frequent symptoms [1,3], we aimed to register these symptoms in a post-COVID-19 symptom questionnaire. Furthermore, we selected standardized and validated health scales to cover these symptoms (cognition, dyspnea, fatigue) along with scales for evaluation of general health (reduced function and quality of life). 

The selection of symptoms (Appendix A) and validated scales were discussed during meetings with specialists from psychiatry, neurology, pulmonology, occupational therapy and infectious diseases, while developing the clinical set up for examination of these patients. The aim was to support clinical assessment, as well as register symptoms and their severity, to be able to characterize the patients and follow the course of disease. The selection of evaluation tools was based on the need for a screening instrument that the patients were able to fill in. Moreover, we needed a broad screening, as the nature of long COVID was unknown. As well as registering a number of symptoms, we wanted to use standardized and previously validated scales for use in patients with persistent symptoms [20,21,22]. If possible, we chose standardized scales already used in other post-COVID cohorts to facilitate comparison [23,24]. We used scales that were possible to repeat at follow-up, and, when possible, we selected scales available in different languages. In the following section the selection of acute symptoms, symptoms in the PCQ and in the validated standardized health scales are described.

Organ-specific and general symptom assessment: Selection of acute and long-term symptoms was based on initial interviews of hospitalized patients [3,11], clinical experience [15,25] and the available literature on acute and long-term symptoms [1,3,11]. This led to the registration of organ-specific and general symptoms. We chose 24 symptoms for retrospective assessment of the acute phase of COVID-19 and 32 symptoms for registration at the first consultation in the Post-COVID Clinic. For each symptom, we registered whether the symptom was new or known before time of infection. 

Cognitive assessment: During the start of the pandemic, we registered cognitive impairment by telephone interviews [3,11]. For this purpose, we selected the Orientation–Memory–Concentration (OMC) test [26], which we continued to use in the Post-COVID Clinic. No cognitive measures had been validated for long COVID patients and other frequently used tests required physical attendance or had too low sensitivity in younger patients. The OMC test was also chosen due to validation of retesting [27]. Further, concentration difficulties and memory problems were included in the PCQ to identify cognitive symptoms and their severity.

Dyspnea assessment: A previously well-validated standardized scale was chosen for the registration of how dyspnea affected level of functioning. The Medical Research Council (MRC) Dyspnea Scale is commonly used for patients with chronic dyspnea [28]. Dyspnea was also evaluated in the PCQ asking whether dyspnea was a problem at rest and/or during physical activity.

Fatigue assessment: As fatigue was a common manifestation in long COVID patients and the mental or physical component was unknown, we used the Fatigue Assessment Scale (FAS) [22], which includes questions on mental as well as physical fatigue and which has previously been used in patients with chronic diseases with multiple symptoms [21]. The advantage of using FAS was also that we were able to register fatigue during the last month, while other questionnaires use a time interval of 6 or 12 months, which would often exceed the onset of the disease. In the PCQ we included concentration difficulties and physical exhaustion to cover mental and physical fatigue, respectively. 

General health assessment: Health-related quality of life and functional disability were recognized markers of morbidity and health outcomes [23,24,29]. The Post-COVID-19 Functional Status Scale (PCFS) was not validated when planning the study, but it was early introduced in the long COVID literature and we found it useful and easy to use. We also included the scale to be able to compare our cohort to other cohorts [24]. For the assessment of health-related quality of life, we used a non-disease specific instrument, the EuroQol visual analogue scale (EQ VAS) and the EuroQol five-dimension five-level (EQ-5D-5L-index) [23], which are well-validated scales used in some of the first long COVID publications [2].

Mental health assessment: To rule out psychiatric illness, we wished to register anxiety, as well as depression and general strain. For this purpose, we used the Symptom Checklist (CMDQ, SCL-13), in which all three conditions could be registered using 13 questions [20,30]. We further screened to rule out bodily distress (BDS) using the 25-item BDS [31].

Background information: Data on demographic characteristics and medical history in the post-COVID cohort were retrieved from the patient file. An interview guide was used by the physician including the variables shown in Table 1 to ensure systematic collection of data.

### 2.2. Setting

The post-COVID clinics in Central Denmark Region were established 1st of February 2021 and located at two hospitals. One in the Department of Infectious Diseases at Aarhus University Hospital (AUH), which covers an uptake area of 1 million people, equivalent to ¾ of the population in the region [32]. In this area, 50,342 individuals had a positive SARS-CoV-2 PCR test in the time period from the beginning of the COVID-19 pandemic in March 2020 until 8th of Aug 2021—i.e., 12 weeks before the end of the study period on the 1st of November 2021, which was predefined as the evaluation point [33]. According to registrations at AUH, a total of 547 patients were evaluated in the Post-COVID Clinic and diagnosed with sequelae of COVID-19 (DB948A) between February and November 2021. This was equivalent to 1.1% of individuals in the hospital uptake area with a positive SARS-CoV-2 PCR test during the study period (547 of 50,342 individuals).

Patients aged 18 and over experiencing unexpected or complex and prolonged symptoms for more than 12 weeks were eligible for referral by their GP. Persistent symptoms were to have occurred in timely correlation after COVID-19 infection and we recommended a screening for other concurrent conditions before referral. Referral letters were declined in case of obvious referral for other reasons, such as referrals on suspicion of adverse effects of vaccination. Before March 2021, the Danish Health Authority recommended patients to be referred as early as six weeks following SARS-CoV-2 infection. However, during February and March 2021, patients were not assessed until three months after onset of symptoms due to a waiting list. Patients on the waiting list were phoned before they were scheduled for assessment. Patients who improved spontaneously during the 12 weeks after onset of symptoms were thus not assessed in the clinic. According to guidelines from the Danish Health Authority, patients with simple or spontaneously improving health issues following COVID-19 were treated by a GP [15]. Patients discharged following hospitalization with COVID-19 were not followed in the clinic unless referred by their GP due to persistent symptoms. In order to document the correlation between infection and SARS-CoV-2, a positive SARS-CoV-2 PCR test or a positive SARS-CoV-2 antibody test were required. No patients in the present cohort diagnosed on the basis of antibody results had a COVID-19 vaccination before the antibody test. In spring 2020, SARS-CoV-2 tests were not available for all. Thus, some patients were seen without a positive PCR or antibody test result (Table 1). In these cases, a clinical diagnosis was based on history of an obvious source and/or symptoms of COVID-19. 

Patients were included in the post-COVID cohort at their first consultation in the Post-COVID Clinic, where they underwent a thorough medical examination by a multidisciplinary team using PCQ/standardized health scales as a part of the examination and assessment. Objective examination, broad biochemical analyses and other paraclinical examination depending on indications, such as lung function test in patients with dyspnea, were performed. 

### 2.3. Definition of Variables for Analysis

Data on demography, medical history and results from paraclinical investigations, along with patient reported data from the PCQ and standardized health scales (Appendix A), were registered in a secured REDCap database [34,35]. Time from onset of symptoms until evaluation in the Post-COVID Clinic were calculated from the date of symptom onset and the date of assessment in the clinic. In case the date of symptom onset was missing, date of positive PCR was used. Data on comorbidities for calculation of Charlson comorbidity index (CCI) [36,37], previous depression and previous stressful episodes were registered by the physician. 

Acute symptoms: Symptoms during the acute phase of COVID-19 were registered as present or not present. In case of registration of “unchanged compared to before the acute phase of COVID-19” or “don’t know”, the response was categorized as “not present” in analyses. 

Long COVID symptoms: The severity of each symptom during a four-week time period before the evaluation in the clinic was graded between 0 and 4 (none, a little, some, a lot, very much). Symptoms “unchanged compared to before the acute phase of COVID-19” were counted as none. In binary analyses, symptoms were considered present in case the answer was some, a lot or very much. A PCQ score was calculated by adding all 31 scores of long COVID symptoms (0–124), as subjective fever was reported as yes or no (0, 1), and thus, not included in the PCQ score. According to COS recommendations [17], which recommend central nervous system (CNS) and cardiovascular (CP) symptoms as outcomes, we calculated organ-specific severity scores from symptoms shown to be more prevalent after COVID-19 [1,4,5,6,7,8,9]. The CNS score was calculated from headache, dizziness, short-term memory problems, concentration difficulties and paresthesia (score 0–20) and the CP score from dyspnea at rest, dyspnea during exercise, cough, chest pain and palpitations (score 0–20). Musculoskeletal (MS) score was also calculated due to considerations of a possible need for inclusion in the core outcome set. MS score was calculated from the sum of joint pain, joint swelling, myalgia, muscle exhaustion and physical fatigue (score 0–20). To investigate associations with symptoms not considered as part of the core outcomes, we further calculated a smell and taste score from altered smell and taste (ST) (score 0–8) and a gastrointestinal (GI) score as the sum of loss of appetite, alternating stool habits, diarrhea, nausea and abdominal pain (score 0–20). 

Standardized scores: The MRC Dyspnea Scale is a 5-point scale 1–5, where 3–5 scores are considered severe [38]. An OMC test result ≤24 was considered impaired [26]. PCFS is a 5-point scale 0–4, where 3–4 scores are considered moderate to severe [24]. Fatigue was assessed by the FAS 10–50, scores ≥ 35 indicate extreme fatigue [39]. The SCL-13 contains items 1–4 assessing anxiety (>5 considered positive), items 2–9 assess general strain on a continuous scale and items 8–13 assess signs of depression (>8 considered positive) [30]. BDS was considered present if 4–5 domains ≥ 4 [31]. The EQ-VAS was registered on a scale of 0–100, and EQ-5D-5L results were compared to population index normal values [23]. 

### 2.4. Data Analyses

Data were analyzed using Stata Intercooled version 17. Median (interquartile range (IQR)), means (standard deviation (sd)) and frequencies were reported. Missing values in PCQ and organ-specific and FAS scores were counted as 0 unless all answers were missing. Symptom scores were compared in two-way graphs with linear prediction of correlation, and statistical significance of correlation was calculated. Spearman correlation coefficients and *p*-values using Bonferroni correction were presented.

## 3. Results

### 3.1. Demographic Characteristics 

Of the 547 patients, 447 (82%) accepted participation in the post-COVID cohort study (Table 1). Participants were evaluated at a median of 6.3 (IQR 4.4–9.9) months after onset of symptoms, had a median age of 47 (IQR 36–56) years and of the patients, 72% were female and 8% were of ethnic origin other than Danish. Twelve percent of patients were hospitalized during the acute phase of COVID-19, 15% of the patients had a CCI above 1 and 13% reported previous episodes of depression. BMI was above 25 in 65% of patients and above 30 in 40%. Five percent of the patients reported excessive alcohol consumption, while 7% of the patients were current smokers. A total of 98% of patients had a positive SARS-CoV-2 PCR or antibody test. Fifty percent of patients had a medium- or long-cycle higher education, of whom 86% were in the workforce, and 8% of the patients lived alone. Of the total 547 registered patients, 72% were female and had a median age of 48, which was similar to the 447 included patients. 

### 3.2. Characteristics of Symptoms

Patients reported several new onset symptoms. During the acute phase of COVID-19, the most reported new onset symptoms were physical exhaustion (94%), headache (82%), myalgia (80%), cough (76%), fever (74%) and altered sense of smell (76%) and taste (73%) (Table 2). Less than 1% of the patients reported not having had any of the symptoms assessed in this study during the acute phase. The most prevalent long COVID symptoms (new onset symptoms of severity some to very much) were disturbed sleep (84%), physical exhaustion (82%), concentration difficulties (82%), short-term memory problems (64%), headache (62%), dyspnea at physical activity (61%), muscle exhaustion (55%) and myalgia (48%). The median of organ-specific and total PCQ score can be seen in Table 2.

Standardized scales indicated impaired cognition in 26% (OMC ≤ 24), severe disability due to dyspnea in 28% (MRC 3–5), functional disability in 33% (PCFS 3–4), extreme fatigue in 62% (FAS ≥ 35) and indication of BDS was found in 11% (4–5 domains ≥ 4) of the patients. The patients rated their health on an EQ VAS scale (1–100) at a mean of 58 (sd 19) and had a mean quality of life according to the EQ-5D-5L index of 0.72 (sd 0.19). Fifteen percent had a depression score of SCL >8 and 28 % had an anxiety score SCL > 5 (Table 2).

Comparing patients hospitalized during the acute phase of COVID-19 with non-hospitalized patients, odds were higher of long COVID dyspnea (odds ratio of dyspnea at physical activity was 1.40 (1.09–1.79) and odds ratio for MRC ≥ 3 was 3.19 (1.64–6.22)). Otherwise, long COVID symptoms did not differ significantly when comparing hospitalized with non-hospitalized patients.

### 3.3. PCQ Score

#### 3.3.1. PCQ and Health Scores

The median PCQ score was 32 (IQR 23–44) (Table 2) and the PCQ score was significantly correlated with all validated health scores (Figure 1). Standardized scales for dyspnea (MRC), functional disability (PCFS) and fatigue (FAS) were significantly positively correlated with the PCQ score; and cognition (OMC), health scale (EQ VAS) and quality of life (EQ-5D-5L) were statistically significantly negatively correlated with the PCQ score. The *p*-value was < 0.0001 for all correlations. The EQ-5D-5L had the highest correlation coefficient of 0.45 (Figure 1).

#### 3.3.2. Organ-Specific Symptom Scores 

We exploratively examined the correlation between the organ-specific symptom scores of CNS, CP, GI and MS symptoms (i.e., organ-specific measures suggested as COS outcomes [17]) and found that these were highly significantly correlated (*p* < 0.000, Figure 2a,b,d,e,g,h), while the smell and taste symptom score did not correlate with organ-specific scores (Figure 2c,f,i). To further explore the standardized health scores, all standardized health scale scores were compared and significantly intercorrelated and each of the five domains of the EQ-5D-5L was compared with PCQ, PCFS, FAS and EQ VAS and all were significantly intercorrelated (*p* < 0.001 for all correlations except EQ-5D-5L anxiety item vs. PCFS).

## 4. Discussion

A total of 1.1% of SARS-CoV-2 positive individuals from the hospital catchment area were referred to the Post-COVID Clinic. The patients were previously healthy and had severely affected health at time of referral. Patients reported multiple symptoms; above 80% of them reported concentration difficulties and physical fatigue, two thirds reported extreme fatigue and one third reported reduced level of functioning 6.3 months after onset of symptoms. The PCQ score correlated with PCFS, FAS and quality of life scores (EQ VAS, EQ-5D-5L), and we suggest the PCQ score to be used as a measure of the individual burden of symptoms.

### 4.1. Assessment of Long COVID 

#### 4.1.1. Organ-Specific and General Symptom Registration

Despite few comorbidities and previous health risk factors, new onset symptoms were reported by more than 80% of patients with respect to disturbed sleep, concentration difficulties and physical exhaustion. Several other symptoms were common (headache, short term memory problems, dyspnea and muscle exhaustion were seen in over 50% of patients) (Table 1). In systematic reviews and meta-analyses of long COVID, the most prevalent reported symptoms are fatigue and “fatigue or muscle weakness”, concentration difficulties [5,6,40,41], headache, dyspnea [5,40,41], memory problems and insomnia [5]. These symptoms were also the most prevalently registered in our cohort, but at much higher percentages for each symptom, with, e.g., dyspnea in 61% in our cohort vs. 14% in the meta-analysis by Chen et al. [5], suggesting that patients referred to the Post-COVID Clinic were severely affected long COVID patients.

#### 4.1.2. Cognitive Assessment

Using the Montreal Cognitive Assessment, cognitive impairment was registered in 17% of hospitalized patients in a UK multicenter study (PHOS-COVID) [42] and cognitive deficits were also detected in non-hospitalized individuals using the Great British Intelligence test [43]. Various other tests were used to document cognitive impairment (SCIP, TRAIL-B, CFQ, TICS-M) [44,45]. Thus, our findings of cognitive deficits in 24% of the patients using the OMC, and concentration difficulties in 82% of the patients according to the PCQ, were in agreement with previous findings. However, the severity, duration and nature of cognitive deficits still need further clarification.

#### 4.1.3. Dyspnea Assessment

Severely affected functioning due to dyspnea (MRC dyspnea score 3–5) was found in 28% of our patients, even though asthma and COPD were only reported in 14% and 2% of the patients, respectively, and only 12% were hospitalized during the acute phase of COVID-19 and had more severe long COVID dyspnea. Huang et al. reported an mMRC ≥ 1 in 26% of patients six months after hospitalization (equal to MRC dyspnea score ≥ 2 in our study). Thus, patients in our cohort reported dyspnea more frequently than hospitalized patients six months after onset of symptoms [10]. COPD patients with an MRC dyspnea score of 3–5 are recommended to be offered rehabilitation [46], which would presumably also be necessary in case of an MRC 3–5 in long COVID patients. Currently, respiratory therapy is generally recommended for long COVID [16], but the criteria for treatment and evaluation of effect have not been established.

#### 4.1.4. Fatigue Assessment

Fatigue is the most prevalently reported symptom in long COVID [4,5,6,8,11,13]. The FAS score has been validated in a number of chronic conditions with extreme fatigue in up to 25% of patients with sarcoidosis [21,47]. Extreme fatigue was also reported in a substantial number of participants (40%) in an online survey on long COVID [48]. In our patient cohort, 62% and 82% reported extreme fatigue and physical exhaustion, respectively, indicating that fatigue is perceived as a very common and debilitating problem for these patients.

#### 4.1.5. General Health Assessment

The PCFS was now validated against EQ-5D-5L and SF-36 [49,50], and in the study by Machado et al., 56% patients, primarily non-hospitalized, had a PCFS of 3–4 at 79 days from the onset of COVID-19 symptoms [50]. This finding was in agreement with our study, in that 33% of the patients reported moderate to severe (3–4) functional impairment when using the PCFS. 

We found an EQ-5D-5L index of 0.72 among our patients, which is a better health-related quality of life score compared with most published data on chronic diseases (−0.20–1.00) [51], but a lower quality of life than the Danish population mean (The mean EQ-5D-5L index in the Danish population is 0.93 at age 20–29 years and 0.83 at age 70–79 years [52]). Activity and pain/dysfunction domains were mainly affected (Table 2). Impaired health documented using EQ-5D-5L has been reported in multiple long COVID studies [6] and the EQ VAS, as well as the EQ-5D-5L index, were also shown to be below pre-infection levels in long COVID patients [42].

In conclusion, assessment of patients in a post-COVID clinic using graded symptom questions in the PCQ and standardized health scales showed that patients were affected in all areas suggested in COS recommendations [17] and standardized health scores indicated that these patients were severely affected and had a high burden of disease.

### 4.2. Evaluation of the PCQ Score

We found that the PCQ score correlated with all standardized health scales. Construct validity evaluation of PCFS was previously found to correlate with long COVID symptoms (and EQ-5D-5L, notably the activity domain) [50], and was also suggested to be used for screening patients in need of careful follow-up and as an outcome in clinical trials [49,50]. Exploring our results in the light of these, our PCQ correlated significantly with all domains in the EQ-5D-5L. We conclude that the PCQ score can be used for evaluating the severity of long COVID symptoms and reflect severely affected health. In a clinical setting, the evaluation of severity using a symptom score is convenient, as an anamnesis with registration of symptoms is part of the clinical assessment. Interestingly, a recent German study suggested a post-COVID severity score based on 12 long COVID complexes. Although development of the score was based on population data, 11 of the 12 complexes were represented in our PCQ. Similar to our study, they found a significant correlation with the EQ-5D-5L Index and EQ VAS [53]. Follow-up studies are needed to determine whether PCQ can be used as a screening tool to predict outcomes. Our PCQ median score was 32 on a scale from 0–124, but a PCQ cut off value for severe long COVID has not been defined.

Exploratively, we constructed organ-specific scores, which were in agreement with NICE, UK guidelines and COS. Only in NICE guidelines, are musculoskeletal symptoms categorized separately, as in this study [16,17,19]. We found all organ-specific symptom scores correlated positively, except for smell and taste symptoms (Figure 2). Huang et al. clustered symptoms to define different phenotypes of long COVID [54] and Canas et al. found three clusters of symptoms suggesting a different pathophysiology [55]. However, different types of long COVID did not seem to be the case in patients referred with more severe long COVID in this study. Severe long COVID seems to consist of CNS, CP, GI as well as MS symptoms. Affected cognition and memory have been suggested to arise from a direct effect through the olfactory bulb [56], however, in our data more severe organ-specific symptoms did not include more severe taste and smell alterations. We suggest that long COVID is a systemic disease with a common underlying pathophysiological mechanism with varying severity. We previously detected myopathy in electrophysiological investigations in 11 of 20 long COVID patients seven months after SARS-CoV-2 infection [57], as well as pathology in muscle biopsies [58], which may be one of the clues to explain this systemic disease.

### 4.3. Demographic Characteristics and Generalizability

Patients in this study were mainly female (72%). An equal number of females and males were infected with SARS-CoV-2 during the pandemic [59]; however, in accordance with our study, multiple studies of long COVID report a consistent and marked overrepresentation of women [4,5,8,9,10,12], though not yet offering an explanation whether this is due to psychosocial or pathophysiologic factors. The consistency across study designs and female preponderance across continents [60] speak to the latter. Although only 1% of the patients did not experience acute symptoms, no more than 12% experienced moderate or severe acute COVID-19 defined by hospitalization. Patients in this study had few comorbidities (15% had CCI above 1), rarely smoked (7%) or drank excessive amounts of alcohol (5%) compared with a high prevalence of comorbidity in hospitalized COVID-19 patients [10] and compared with the more frequently occurring health risk factors in the Danish population [61]. We found moderate obesity in 65% and severe obesity 40%, compared to 61% and 23%, respectively, in the age group 45–55 in Danish data from 2021 [61]. High BMI was previously described as a predictive factor for long COVID [62]. However, it seems likely that long COVID disease causes an increase in weight, as patients report difficulties in performing usual activities and sports [16]. Psychiatric diseases were reported to be more frequent following COVID-19 [63]. This was not the case in our cohort. Previous depression was registered in 13% of the patients and 15% of the patients registered signs of depression, which seemed to be at the same level as the background population, in which 14% report current depression [61]. Further, we found indication of bodily distress in only 11%, which was less than in the background population [64]. Higher education in 50% of patients in this study and employment in 86% (compared to 1 out of 5 individuals outside the workforce in Denmark in 2021, according to Statistics Denmark), may indicate that patients referred to the clinic were of a higher socioeconomic status, leaving the question of possible inequality of access to health services, as transmission of SARS-CoV-2 undoubtedly was just as frequent in lower socioeconomic residential areas and jobs [65].

Returning back to normal life after infection has previously been shown to exacerbate even subtle cognitive symptoms [66], however, collectively, we found no evidence of predisposing health risk factors or psychosocial risk factors.

#### 4.3.1. Limitations

Multiple factors may affect the selection of patients referred to the Post-COVID Clinic, such as the GP or patients’ suspicion of long COVID, knowledge of the existence of a post-COVID clinic, level of health literacy or the presence of symptoms affecting daily functioning. Patients were primarily women, middle aged, often of higher education, which may affect patient-reported outcomes. Recall bias may also have affected the registration of symptoms during acute COVID-19. While interviewer bias may have caused the registration of more symptoms while filling in a questionnaire used in the clinical assessment, we have no reason to think this would have caused differences in registration in the PCQ vs. standardized health scores. Missing values considered as zero and “unchanged symptom” considered not present might have underestimated severity of disease, but our cohort, however, still appeared to have numerous symptoms. In this study, we use the unique opportunity to describe some of the more severely affected long COVID patients referred due to unexpected, complex and prolonged symptoms. The participation rate was high (82%) and non-participants’ age and gender did not differ from participants. We registered new onset symptoms and SARS-CoV-2 test results to ensure the condition described was, in fact, long COVID.

#### 4.3.2. Generalizability

Collectively, results indicate that our patient cohort included patients who were previously healthy and who were among some of the more severely affected long COVID patients. Post-COVID clinics in other regions of Denmark used the same referral criteria, and a number of post-COVID clinics worldwide report seeing patients with multiple symptoms and low functional level [13,14,15,16,17]. These clinics describe patients with similar demographic characteristics, and our data could thus be generalizable to long COVID patients in these settings. Health seeking behavior and doctors’ knowledge about the post-COVID clinic may have affected the inclusion of patients and we do not claim that our cohort include all severely affected long COVID patients.

No specific diagnostic or paraclinical tests are suggested in national or international guidelines, and the various assessment tools used in different settings underline the complexity of the numerous symptoms described in patients with long COVID. We did not aim to cover the complexity of long COVID-19 disease, and the PCQ was prepared as a supplement to the anamnesis. Other studies registered up to 115 different symptoms and the registration of more symptoms might have detected overlooked phenomena [67]. Even so, the PCQ described the presence of all of the more frequently reported long COVID symptoms in other studies as discussed above.

The symptoms reported in the PCQ occur after other infectious diseases, but little is known of the prevalence and duration. Future studies might benefit from focusing on the systematic collection of data following epidemics.

## 5. Conclusions

The post-COVID clinics were established to receive patients with persistent symptoms after COVID-19 infection with regard to diagnostics and treatment. This made it possible to create a cohort of patients with severe symptoms. Multiple symptoms and extreme fatigue, as well as reduced functional level, were reported. A PCQ questionnaire including a wide spectrum of symptoms significantly correlated with burden of disease in standardized health scales and may be used for evaluation of severity of symptoms in long COVID. Follow-up studies are needed to evaluate whether symptom scales can predict recovery. We hope this study will encourage the development of common validated screening and outcome measures for use in prospective cohort studies and treatment trials.

## Figures and Tables

**Figure 1 jcm-11-07338-f001:**
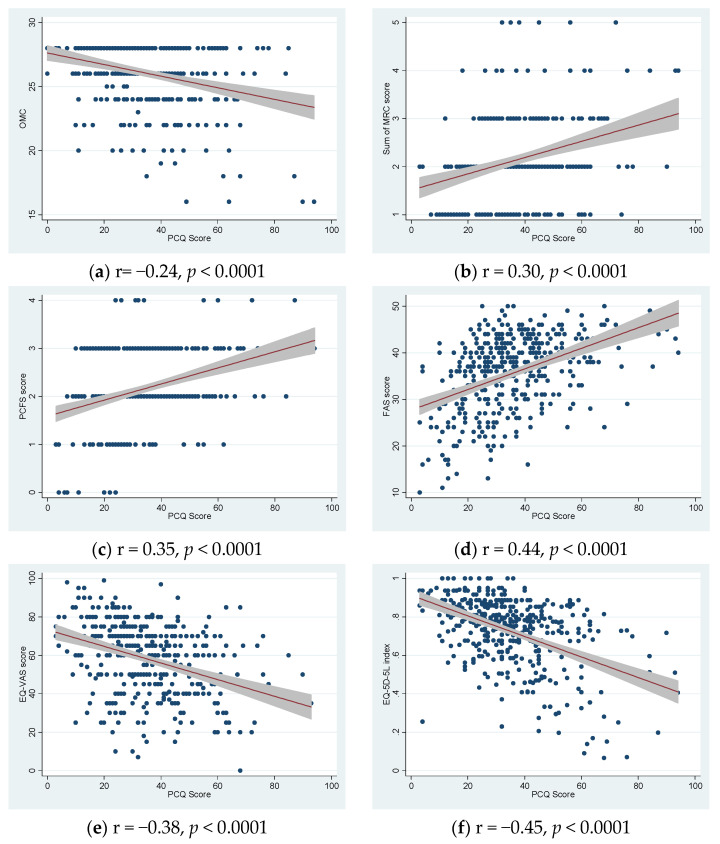
Correlation plots comparing post-COVID symptom questionnaire (PCQ) with validated health scores ((**a**) OMC, (**b**) MRC dyspnea, (**c**) PCFS, (**d**) FAS, (**e**) EQ-VAS and (**f**) EQ-5D-5L index). PCQ Score: sum of all 31 registered symptom scores (0–124). OMC: Orientation Concentration Memory test (0–28). MRC: Medical Research Council dyspnea score (1–5). PCFS: Post-COVID Functional Status Scale (0–4). FAS: Fatigue Assessment Scale (10–50). Health-related quality of life: EQ-5D-5L index and EQ VAS (0–100). r = Spearman correlation coefficient (rho).

**Figure 2 jcm-11-07338-f002:**
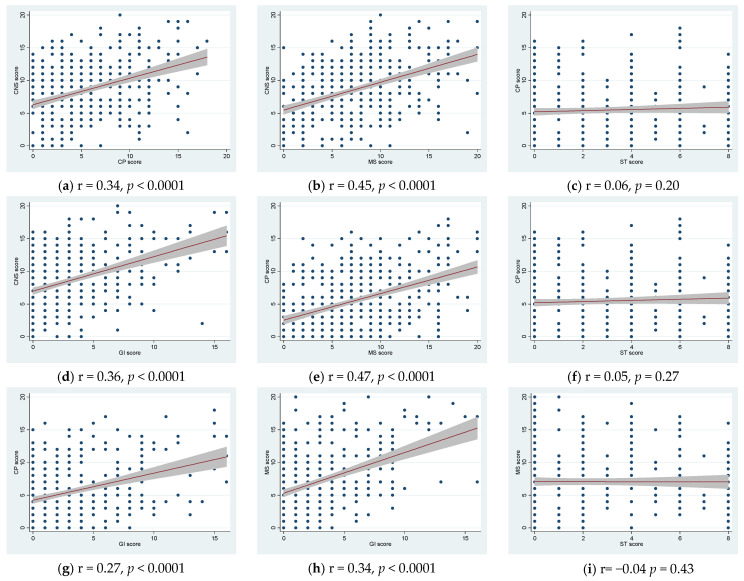
Correlation plots comparing organ-specific symptom scores. CNS score (central nervous system score): sum of headache, dizziness, short term memory problems, concentration difficulties and paresthesia scores (0–20). CP score (cardiopulmonary score): dyspnea at rest, dyspnea at exercise, cough, chest pain, palpitations (0–20). MS score (musculoskeletal score): sum of joint pain, joint swelling, myalgia, muscle exhaustion, physical fatigue scores (0–20). GI score (gastrointestinal score): sum of loss of appetite, alternating stool habits, diarrhea, nausea and abdominal pain scores (0–20). ST (smell and taste score: altered smell and altered taste scores (0–8). (**a**–**d**) CNS vs. CP, MS, ST and GI scores. (**e**–**g**) CP vs. MS, ST and GI score. (**h**,**i**) MS vs. GI and ST score. Bonferroni correction *p* < 0.006.

**Table 1 jcm-11-07338-t001:** Characteristics of patients in a Post COVID Clinic.

	% ^1^	n/N ^2^
Sex (male)	28	127/447
Age in years	Median 47	IQR 36–56
Other ethnic origin than Danish	8	36/446
Familiar dispositions	41	183/447
Autoimmune	12	52/447
Metabolic	23	104/447
Allergic	16	72/447
Transmission of SARS-CoV-2		
Work	37	166/447
Family	26	116/447
Travel	3	14/447
Unknown	18	80/447
Other	14	63/447
Time from symptom onset, median	6.3	IQR 4.4–9.9
Positive SARS-CoV-2 PCR test	87	391/447
Positive SARS-CoV-2 Ab test at follow-up	77	345/447
No positive SARS-CoV-2 test	2	10/447
Hospitalized ^3^	12	53/447
Charlson comorbidity index >1	15	68/447
Diabetes	2	7/447
Asthma	14	63/447
COPD	2	8/447
Coronary heart disease	1	5/447
Cerebrovascular disease	<1	–
Hypertension	10	46/447
Malignancy (previous)	3	15/447
Connective tissue disease	1	5/447
Immunodeficiency	<1	–
Previous depression	13	56/447
Previous stressful episode	12	54/447
Medicine		
ACE or AT2 rec inhibitor	11	45/425
Statins	10	41/424
Steroids	1	5/421
Other immunosuppressant	2	9/422
Current smoker	7	30/412
Alcohol, > 7 units per week	5	20/404
BMI > 25	65	266/412
Socioeconomic status		
Education		
Primary school	9	42/447
High school	15	66/447
Vocational education	24	109/447
Medium-cycle higher education	40	178/447
Long-cycle higher education	10	43/447
Work		
Employed	81	361/447
Self-employed	5	24/447
Voluntary	2	8/447
Student	5	22/447
Stay at home	2	10/447
Retired	6	25/447
Not living alone	82	300/364
Size of housing, mean m^2^	143	sd 64

^1^ Data reported as % unless otherwise stated. IQR: inter quartile range. sd: standard deviation. ^2^ Data missing in patient file shown as N less than 447. ^3^ Hospitalized in the acute phase of COVID-19.

**Table 2 jcm-11-07338-t002:** Patient-reported new onset symptoms and standardized health scale results in patients in a post-COVID clinic.

	Symptoms -Acute Phase of COVID-19 ^1,3^		Long COVID Symptoms (Some to Very Much) ^2,3^	
	%	n/N	%	n/N
Central Nervous system				
Headache	82	358/434	62	262/423
Dizziness	67	283/421	37	156/422
Paresthesia	38	154/408	27	114/417
Confusion	31	128/413	–	–
Concentration difficulties	–	–	82	337/413
Short-term memory problems	–	–	64	268/422
Long-term memory problems	–	–	43	179/420
Disturbed smell and taste				
Impaired smell	76	323/425	37	156/424
Impaired taste	73	311/425	34	143/422
Upper respiratory tract				
Runny nose or nasal obstruction	52	218/417	23	95/417
Sore throat	57	242/427	15	61/416
Cough	76	329/434	17	71/415
Expectoration	32	136/419	12	49/420
Cardiopulmonary				
Dyspnea at rest	–	–	29	123/423
Dyspnea at physical activity	73	314/431	61	260/425
Chest pain	53	225/423	28	117/422
Palpitations	48	202/419	30	126/422
Gastrointestinal				
Loss of appetite	–	–	18	71/390
Nausea	47	199/422	17	71/421
Diarrhea	31	131/418	10	41/416
Abdominal pain	27	113/413	13	56/417
Altered bowel habits	–	–	27	112/411
Dermatology				
Skin rash	13	55/411	8	34/408
Itching skin	19	80/411	15	64/415
Musculoskeletal				
Joint pain	68	288/424	41	168/414
Swollen joints	13	52/410	10	39/406
Myalgia	80	340/432	48	200/416
Muscle exhaustion	–	–	55	224/407
Physical exhaustion	94	398/425	82	343/420
General symptoms				
Subjective fever	74	322/434	13	48/373
Duration of fever > 7 days	55	115/211	–	–
Weight loss during initial illness	32	134/417	–	–
Disturbed sleep	–	–	84	357/423
Problems falling asleep	–	–	40	172/425
Awakening	–	–	56	235/419
Symptom and health scores				
PCQ total score	–	–	median 32	IQR 23–44
CNS score	–	–	median 9	IQR 5–11
ST score	–	–	median 5	IQR 2–8
CP score	–	–	median 1	IQR 0–4
GI score	–	–	median 2	IQR 0–4
Musculoskeletal score	–	–	median 7	IQR 3–10
Orientation-concentration-memory OMC ≤ 24	–	–	26	94/366
Dyspnea score, MRC ≥ 3	–	–	28	94/336
Post COVID Functional Status, PCFS 3–4	–	–	33	128/387
Fatigue Assessment Scale, FAS ≥ 35	–	–	62	244/391
Depression score > 8, SCL-13 item 8–13	–	–	15	62/414
Anxiety score > 5, SCL-13, item 1–4	–	–	28	116/417
General strain, SCL-13, item 2–9	–	–	mean 14	sd 6.3
Bodily Distress Syndrome (BDS)	–	–	11	39/355
EQ VAS 1–100 ^4^	–	–	60	IQR 45–70
EQ-5D-5L index ^4^	–	–	0.72	sd 0.19
EQ-5D-5L items	–	–	–	–
Mobility 3–5	–	–	14	60/421
Self-care 3–5	–	–	4	19/424
Usual activity 3–5	–	–	58	244/422
Pain/discomfort 3–5	–	–	60	254/423
Anxiety/depression 3–5	–	–	18	75/420

^1^ Acute symptoms: the answer “don’t know” considered as not present (42 cases of palpitations, 54 cases of swollen joints and in 1–24 cases of the other symptoms), only new onset symptoms counted. ^2^ Long COVID symptoms: counted as present in case of answer “some”, “a lot” or “very much”; only new onset symptoms are counted. ^3^ Data reported as % unless otherwise stated. IQR: inter quartile range. sd: standard deviation. ^4^ Higher scores indicate better health. Missing values in symptom scores and FAS were counted as 0 unless all answers were missing.

## Data Availability

Data were registered in a secured REDCap database at https://redcap.au.dk/ accessed on 14 March 2022. The data are not publicly available due to ethical restrictions.

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
