# Peer review of "Characteristics of a Danish Post-COVID Cohort Referred for Examination due to Persistent Symptoms Six Months after Mild Acute COVID-19"

_jcm, 2022, doi:10.3390/jcm11247338_

Round 1

Reviewer 1 Report

The present study thoroughly describes a cohort through employment of a number of validated clinical assessment scores including cognitive assessment, organ specific symptoms, graduation of breathing difficulties/dyspnea, fatigue, and quality of life.

The study provides a baseline for burden of symptoms in a large cohort from an industrialized high-income country.  

The manuscript is extensive but well-written and does, in this reviewer’s opinion, provide unbiased results/data on a large cohort of primarily healthy middle-aged patients of both genders and as stated above also provide a baseline for other researchers when it comes to clinical assessment scores schemes and the results from this assessment.      

Abstract:

No comments. The size of the municipality should also be stated in the “materials and methods”

Introduction:

Well-written. Could the authors mention the headline in the slightly varying definitions of long COVID from WHO, US and NICE (difference in time after COVID and symptoms).  

Material, methods, statistics:

Well-described in accordance with the results presented. Size of municipality and eventual percentage of the Danish population in the first section.

Were the only criteria for referral to the clinic a positive COVID test or antibody test and lack of recovery after 12 weeks? Were certain patients declined a visit to the clinic i.e. based on disease history or competing explanations for as depression?   

Were any routine clinical assessments performed?

Were lung-function tests performed on routine or only among patients with this specific complaint.

Results:

Were any patients declines referral?

Comparisons between hospitalized and non-hospitalized patients are made in the discussion (lines 339-343) but not in general. Were there any differences between these two groups on any parameter?

Figure 2: Not sure that I can follow the sub-headings under each graph. CNS is missing and smell-and taste score provided 3 times. I have likely misunderstood – clarification is needed.

Aware that this may be too extensive for the purpose of this manuscript. But, was there any associations to no recovering an blood biochemistry, lung function, radiology findings or being hospitalized versus non-hospitalized.

Discussion:

Good outline with division into each symptom category.

Was the burden of disease graded from the multiple scores most applicable to WHO, US or NICE guidelines? 

Figures and tables:

Uncertain -but would division into hospitalized or non hospitalized provide additional information?

Reviewer 2 Report

This is novel and interesting study and result, 

I only concern the missing titles of vertical axis in "every figure" of Figure 1 & 2, and the meaning of the "Arabic number". Please revise above points for clear readness.

Here are my comments:

1. In production, what is the hypothesis of current study?

2. Why the title “…six months after…”, but in line 64 “…first nine months…”? not compatible?

3. In table 1, the “n/N” column, why some N not always 447? Could the authors explain?

Reviewer 3 Report

Title: Characteristics of A Danish Post COVID Cohort Referred for Examination due to Persistent Symptoms Six Months after Mild Acute COVID-19

In this manuscript, described post COVID clinic at the time of diagnosis, they reported multiple symptoms with se- 24 verily affected health. The PCQ score could be used as valid measure of Post-COVID severity.

The manuscript, even though including recent and relevant literature, it should be structured more clearly and follow a line of thought.

1.      What's the hypothesis of this study?

2.      This study was to describe patients included first nine months of the pandemic in a Post COVID Clinic in a Danish region. Why just first nine months?

3.      Taken together, in this study, some people continue to experience symptoms beyond the acute phase of infection. Consequently, long COVID has appeared as a new term in the literature to describe the long term symptoms of a COVID-19 infection. Just describe a few line regarding the history patients.
